# Association of Macronutrients Intake with Body Composition and Sarcopenic Obesity in Children and Adolescents: A Population-Based Analysis of the National Health and Nutrition Examination Survey (NHANES) 2011–2018

**DOI:** 10.3390/nu15102307

**Published:** 2023-05-15

**Authors:** Bing Yang, Chengjun Tang, Zumin Shi, Liwang Gao

**Affiliations:** 1Department of Disinfection and Vector Control, Beijing Chaoyang District Center for Disease Control and Prevention, Beijing 100021, China; 2Human Nutrition Department, College of Health Sciences, QU Health, Qatar University, Doha 2713, Qatar; 3Center for Non-Communicable Disease Management, Beijing Children’s Hospital, Capital Medical University, National Center for Children’s Health, Beijing 100045, China; 4School of Public Health, Capital Medical University, Beijing 100069, China

**Keywords:** macronutrient, body composition, sarcopenic obesity, children and adolescents

## Abstract

The association of macronutrients intake with body composition and sarcopenic obesity remains uncertain in children and adolescents. We aimed to explore the association between macronutrients intake and body composition, especially sarcopenic obesity, in children and adolescents residing in the United States. The study utilized data from 5412 participants aged 6–17 years who attended NHANES between 2011 and 2018. Body composition was assessed using DXA, and nutrient intake was based on 24-h recall. Multivariable linear regression and multinomial logistic regression were used. The unweighted prevalence of sarcopenic obesity was 15.6%. A higher percentage of energy (5 %E) from fat was inversely associated with muscle mass but positively associated with fat mass and sarcopenic obesity. Substituting carbohydrate (5 %E) with fat decreased muscle mass by 0.03 (95% CI 0.01 to 0.06) but increased fat mass by 0.03 (95% CI 0.01 to 0.06) and increased the prevalence of sarcopenic obesity by 254% (95% CI 15% to 487%). Replacing protein intake with fat intake also increased the OR of sarcopenic obesity (OR, 2.36 [95% CI 1.18 to 3.18]). In conclusion, a high-fat diet, coupled with low carbohydrate/protein intake, is associated with sarcopenic obesity among children and adolescents. The change in children’s diet towards a healthy diet with low fat composition may help prevent sarcopenic obesity. However, randomized clinical trials or longitudinal studies are needed to further validate our findings.

## 1. Introduction

Sarcopenia and sarcopenic obesity (SO), defined as the co-occurrence of increased fat mass and sarcopenia, once thought to afflict only the elderly, are now connected to the pediatric population [1]. Based on a systematic review, the prevalence of SO in children and adolescents ranged from 5.7% to 69.7% in girls and between 7.2% and 81.3% in boys [2], and SO has been found to be associated with cardiometabolic outcomes, inflammation, and mental health in children and adults [3].

Macronutrients intake plays an important role in body composition. Previous research on the effects of nutrition on body composition in children and adolescents has mainly focused on the relationship between dietary patterns and body mass index (BMI) and fat mass [4,5]. In a systematic review conducted by the 2020 United States Dietary Guidelines Advisory Committee and Dietary Patterns Subcommittee, it was found that dietary patterns with lower intake of fruits, vegetables, whole grains, and low-fat dairy, and higher intake of added sugars, refined grains, fried potatoes, and processed meats, are associated with higher fat mass index (FMI) and BMI in children and adolescents [6]. Limited studies have investigated the effect of macronutrients distribution on muscle and SO in children, and the results have been inconsistent. Some studies have shown that higher protein intake is associated with increased muscle mass [7,8], while others have found no such association [9]. Since total energy intake varies relatively small within individuals, increasing the consumption of one type of macronutrient means decreasing the intake of others. When protein intake is analyzed separately, the interpretation of the results can become difficult, as the estimated effect of protein may depend on the other macronutrients (carbohydrates and fat) that they replace. This can lead to contradictory research results, and it also makes it unclear what the optimal adjustment strategy for improving the distribution of macronutrients in body composition is.

A balanced consumption of macronutrients, including carbohydrates, proteins, and fats, can help regulate body composition and prevent SO. Isocaloric substitution analysis, wherein one macronutrient is replaced by another while maintaining the same caloric intake, has yielded promising results in altering body composition [10]. As a result, there is growing interest in investigating the relationship between macronutrients intake and body composition in children and adolescents. However, prior research on this subject has been largely limited to small sample sizes or adult populations [11,12], with a dearth of large-scale, nationally representative studies exploring the association between macronutrients intake and body composition, particularly SO, in children and adolescents within the United States.

To address gaps in knowledge, the objective of this study was to conduct a comprehensive examination of the relationship between the distribution of dietary macronutrients and parameters of body composition by using data from National Health and Nutrition Examination Survey (NHANES) 2011–2018, and to investigate whether isocaloric substitution of macronutrients is associated with SO in children and adolescents in United States.

## 2. Materials and Methods

### 2.1. The Study Design and Study Sample

NHANES is a cross-sectional survey in the United States that is designed to assess the health and nutritional status of both children and adults on a national level. The survey is continuous and has been conducted from 1999 to the present day. It employs a stratified multistage probability sample based on the selection of counties, blocks, households, and individuals, in order to accurately represent the civilian, non-institutionalized United States population [13]. The survey data and procedures can be accessed at https://www.cdc.gov/nchs/nhanes/index.htm (accessed on 20 November 2022). For this study, we utilized data from four NHANES cycles spanning from 2011 to 2018, focusing solely on children aged 6–17 years who had complete information regarding their body composition, valid diet recall, and covariates. Of the original 39,156 participants in the 2011–2018 NHANES cycles, 9153 were aged 6–17 years, but 3142 were missing data on Dual Energy X-ray Absorptiometry (DXA) and 481 were missing data on dietary recall. After excluding 118 participants with extreme 1% low or high total energy intake for each age group, a total of 5412 participants remained for inclusion in the final analysis (Figure 1).

NHANES was approved by the US Centers for Disease Control and Prevention/National Center for Health Statistics Ethics Review Board. Written informed consent was obtained from all the participants and/or their parents.

### 2.2. Anthropometric Measurements and Whole-Body DXA Scanning

Height and weight were measured by trained health technicians using standardized protocols, and BMI was calculated. Whole-body DXA scanning was performed to test body composition indicators using Hologic fan-beam densitometers (Hologic, Bedford, MA, USA) by trained technicians according to the International Society of Clinical Densitometry’s (ISCD) standard operating procedure [14,15]. Data from DXA included values for fat mass (g), bone mineral content (g), and lean mass (g) of the whole body.
$$\mathrm{FMI}\;\left(\mathrm{kg}/m^{2}\right)=\mathrm{whole}\;\mathrm{fat}\;\mathrm{mass}/{\mathrm{height}}^{2}$$
$$\mathrm{Muscle}\;\mathrm{mass}\;\mathrm{index}\;\left(\mathrm{MMI},\;\mathrm{kg}/m^{2}\right)=\left(\mathrm{whole}\;\mathrm{lean}\;\mathrm{mass}-\mathrm{bone}\;\mathrm{mineral}\;\mathrm{content}\right)/{\mathrm{height}}^{2}$$
$$\mathrm{Muscle}\;\mathrm{to}\;\mathrm{fat}\;\mathrm{ratio}\;\left(\mathrm{MFR}\right)=\left(\mathrm{whole}\;\mathrm{lean}\;\mathrm{mass}-\mathrm{bone}\;\mathrm{mineral}\;\mathrm{content}\right)/\mathrm{whole}\;\mathrm{fat}\;\mathrm{mass}$$

### 2.3. Assessment of Macronutrients Intake Distribution

The nutrient intake obtained from food was estimated using two non-consecutive 24-h dietary recalls through the automated multiple-pass method in NHANES. All participants underwent in-person interviews for the initial 24-h dietary recall interview. Then, 3–10 days later, a portion of the children and adult participants took part in a second, telephone-based 24-h dietary recall interview. The accuracy of food recalls was enhanced through this method by collecting a self-reported food list, probing for any forgotten foods, obtaining details of foods, and finally, probing for any other food items. Participants also reported the amount of food items using a standard set of measuring guides. Daily total consumption of carbohydrates, protein, fat, and dietary fiber were calculated accordingly. In order to maintain the largest possible sample size and to use more accurate face-to-face survey data, the recalls were chosen from the first day. The recalls in children and adolescents were chosen from the first day. The daily total energy intake was calculated by summing the calories from carbohydrates, protein, and fat, where 1 g of carbohydrate equals 4 kcal, 1 g of protein equals 4 kcal, and 1 g of fat equals 9 kcal [16].

### 2.4. Covariates

For statistical analyses, the following data were further collected: ethnicity, dietary fiber intake, and physical activity. The detailed acquisition process and measuring method of dietary fiber intake are available at www.cdc.gov/nchs/nhanes (accessed on 20 November 2022). Physical activity was assessed by weekly metabolic equivalent task (MET) minute aggregated scores. According to the NHANES recommendations, weekly MET-minutes were calculated as follows: (8.0 MET scores × (weekly minutes of vigorous work-related activity + weekly minutes of vigorous leisure-time physical activity)) + (4.0 MET scores × (weekly minutes of moderate work-related activity + weekly minutes of moderate leisure-time physical activity + weekly minutes of walking or bicycling for transportation)).

### 2.5. Definitions of Sarcopenia and Obesity 

The age- and sex-specific International Obesity Task Force (IOTF) criteria were used to define weight status as normal weight, overweight, and obesity [17]. According to the 2019 ISCD Pediatrics Position Statement and World Health Organization used gender and age-specific Z-score to define children and adolescents’ osteoporosis and nutritional development, and previous studies found that the fat mass Z-score exceeding 1.0 would significantly deepen cardiovascular impact [18], low muscle mass was defined as MMI Z-score less than −1.0, and high fat mass was defined as FMI Z-score more than 1.0. Sarcopenia was defined as MFR ≥ mean MFR-1SD of the third BMI quintile [19]. Based on sarcopenia and obesity, we created a new variable, including (1) normal, (2) sarcopenia alone, (3) obesity alone, and (4) both sarcopenia and obesity.

### 2.6. Statistical Analysis

Continuous variables, such as age and body composition parameters, were expressed as mean with standard deviation (SD), and categorical variables, such as gender and ethnicity, were expressed as the frequency with percentage. The comparisons between the groups were performed using ANOVA tests for continuous variables and χ^2^ tests for categorical variables.

FMI and MMI were standardized to a mean of 0 and a standard deviation of 1 specific for sex and age before analysis, and multiple imputations with chained equations were performed for participants with missing covariates data, assuming data were conditionally missing at random. Multivariate regression analyses were adopted to evaluate the independent association of macronutrients distribution with body composition indicators, multinomial logistic regression was performed to further evaluate the association of macronutrients distribution with SO, and ethnicity was modeled as a random effect in the regression. We constructed two distinct models using a single and a multivariable generalized linear model, including crude model (no covariate was adjusted) and model 1 (age, gender, dietary fiber intake, and total energy intake were adjusted).

Isocaloric substitution analysis was performed to evaluate whether substituting a certain type of macronutrient with another is associated with body composition indicators and SO through the multivariate nutrient density method [20]. Analyses were conducted using Stata software version 14 (StataCorp, College Station, TX, USA). Coefficients with 95% confidence intervals (CI) were recorded. *p* values less than 0.05 (two-sided) were considered statistically significant. 

## 3. Results

### 3.1. General Characteristics of the Participants

Table 1 shows the sample characteristics. Overall, the mean age of the participants was 12.3 years (SD 2.9). The unweighted prevalence of obesity alone, sarcopenia alone, and SO was 2.9%, 11.0%, and 15.6%, respectively. There was a significant difference in age by obesity and sarcopenic status. The lowest age was found among those with sarcopenia alone while the highest age was among those with obesity alone. Energy intake varied by obesity and sarcopenic status, with the highest found in the obesity alone group.

Participants with SO had a significantly higher percentage of energy intake from fat and a lower percentage from carbohydrate than those with a normal body weight. In addition, children with SO had the highest fat mass and nearly the lowest MET. 

### 3.2. Associations of Macronutrients Distribution with Body Composition Indicators and Sarcopenic Obesity

The associations of macronutrients intake with muscle mass and fat mass were shown in Table 2. Carbohydrate intake was negatively correlated with FMI Z-score (β, −0.02 (95% CI, −0.04 to −0.01); *p* = 0.032) and was associated with lower risk of high fat (OR, 0.88 (95% CI, 0.76 to 0.99); *p* = 0.023). On the contrary, fat intake was positively associated with FMI Z-score (β, 0.03 (95% CI, 0.01 to 0.06); *p* = 0.042) and the prevalence of high fat (OR, 1.13 [95% CI, 1.02 to 1.35]; *p* = 0.009). In an unadjusted model, protein intake was positively correlated with MMI Z-score (β, 0.32 [95% CI, 0.24 to 0.40]; *p* < 0.001), and carbohydrate intake was negatively associated with MMI Z-score (β, −0.09 [95% CI, −0.13 to −0.05]; *p* < 0.001). However, the association became insignificant after adjusting for covariates. Nevertheless, fat intake was significantly associated with MMI Z-score (β, −0.03 (95% CI, −0.06 to −0.01); *p* = 0.035) after adjusting for the covariates.

Multinomial logistic regression was performed to further evaluate the association of macronutrients distribution with SO adjusted for covariates in Table 3. In general, higher fat intake was associated with greater odds of obesity alone (OR, 1.10 (95% CI, 1.01 to 1.20); *p* = 0.012), sarcopenia alone (OR, 3.54 (95% CI, 1.14 to 5.08); *p* = 0.024), and SO (OR, 4.42 (95% CI, 1.47 to 6.59); *p* = 0.012). Furthermore, higher carbohydrate intake was associated with lower odds of obesity alone (OR, 0.95 (95% CI, 0.91 to 0.99); *p* = 0.041).

### 3.3. Isocaloric Substitution Analysis Results 

Table 4 and Table 5 show the results of isocaloric macronutrient substitutions. The replacement of 5 %E intake from carbohydrate with isocaloric fat was associated with a higher FMI Z-score (β, 0.03 (95% CI, 0.01 to 0.06); *p* = 0.022) and a lower MMI Z-score (β −0.03 (95% CI −0.06 to −0.01); *p* = 0.013) in Table 4. Substituting 5 %E of carbohydrate intake with fat was associated with increased ORs for both obesity and sarcopenia with OR of 3.66 (95% CI 1.19 to 5.24, *p* = 0.032) for obesity alone, 1.17 (95% CI 1.04 to 1.32, *p* = 0.025) for sarcopenia alone, and 3.54 (95% CI 1.15 to 5.87, *p* = 0.026) for SO. A similarly increased OR was found for the substitution of protein with fat. The corresponding ORs were 2.25 (95% CI 1.53 to 4.40), 1.21 (95% CI 1.07 to 1.52), and 2.36 (95% CI 1.18 to 3.18), respectively.

## 4. Discussion

The comprehensive and representative cross-sectional study, featuring precise body composition parameters and dietary intake data, presented a unique opportunity to conduct this investigation. Our study was conducted among children and adolescents, whose body composition is rapidly developing, especially with the rapid accumulation of muscle and bone during childhood [21,22]. Compared to adult studies, the results of this study also provide important evidence for the intervention and early prevention of unhealthy body composition in children. Our findings indicate that a high intake of fat was associated with a lower MMI Z-scores, higher FMI Z-scores, and an increased prevalence of SO. Conversely, a higher percentage of energy derived from carbohydrate intake was associated with a lower likelihood of having high fat mass and SO. A high intake of protein was positively correlated with muscle mass index. Replacing carbohydrates with fat was associated with a high prevalence of SO.

### 4.1. Comparison with Other Studies

The definition of SO varies between studies, but all of the results show that the child SO burden is serious [2]. A study using DXA that defined SO as the mean value −1 SD of MFR for the third BMI quintile among Korean children found that there was a 24.3% SO prevalence in girls and 32.1% in boys. In addition, the highest prevalence of SO was found in the usage of DXA and the lower 10% of gender-specific appendicular skeletal muscle mass/weight in boys (81.3%), using bioelectrical impedance analysis along with >90th percentile of fat mass/fat-free mass in girls (69.7%) [23,24]. Our results extended these findings to the American children and adolescents: the unweighted prevalence of SO was 15.6%, which was higher than the rate of sarcopenia alone and obesity alone. These findings highlight the crucial importance of body composition control and SO prevention in adolescents in counteracting metabolic outcomes and depression in adulthood.

Previous research has established that excessive consumption of fat leads to an increase in body weight in adults [25,26]. This study has taken it further by revealing that a high intake of fat is linked to an increase in body fat mass and a decrease in muscle mass in children. These findings suggested that fat intake has a significant impact not only on body weight, but also on unhealthy body composition. Moreover, the results of this study demonstrated that substituting carbohydrates with fat can lead to a decrease in muscle mass, which is consistent with a review study suggesting that ketogenic diets have negative and long-term effects on muscle, such as fat accumulation and impaired muscle function [27]. 

A high consumption of processed meat is one of the remarkable features of the western dietary pattern. It is well-known that protein is an indispensable nutrient for muscle metabolism. A randomized trial in healthy adults showed that consuming dietary protein at levels exceeding the recommended dietary allowance may protect fat-free mass during short-term weight loss [28], and a meta-analysis of 24 studies showed that older adults retained more lean mass and lost more fat mass during weight loss when consuming higher protein diets [29]. Furthermore, a prospective cohort involving 3991 children found that protein from plant sources was associated with a higher fat-free mass [30]. However, there are inconsistent findings on the effect of protein on body fat. While many studies suggested that high protein intake reduces body fat [29], a population-based cross-sectional study of 4478 middle-aged and older adults showed that the dietary intake of protein is positively associated with percentage of body fat [31], and a longitudinal study of 3572 children indicated that high protein intake during infancy, especially from animal food sources, was persistently associated with adiposity up to age 10 years [32], which suggested the potential importance of the protein intake as a factor in obesity.

On the other hand, our results showed that a high intake of carbohydrates may contribute to the reduction of body fat mass, and a cross-sectional study conducted on Samoan children also revealed that increased carbohydrate consumption was associated with a lower fat mass percentage and higher lean mass percentage [33]. These findings contradict other studies that have indicated a connection between a high-carbohydrate diet and the subsequent development of obesity and cardiometabolic diseases [34,35]. One potential explanation could be the source of carbohydrate intake among these children. Approximately 60% of those children’s daily calorie intake came from carbohydrates, over half of which came from traditional starchy root crops (taro, breadfruit, bananas, and potatoes) with a lower glycemic index than processed foods [33]. Regarding muscles, we did not find any effects of carbohydrate intake, which is consistent with the previous study in 3573 children aged 1–10 years [36]. In addition, a randomized controlled trial in adults aged 18 to 41 also did not find that the intake of carbohydrates was associated with muscle mass [37]. 

### 4.2. Mechanism of Sarcopenic Obesity Caused by High Fat Diet

The potential mechanism underlying the reduction in muscle mass and increases in obesity risk caused by a high-fat diet is related to energy metabolism, gene expression, and sleep duration [38,39,40,41,42,43]. In ketogenic diet-fed mice, the circulating levels of free fatty acids increased by up to 700%, the plasma corticosterone level increased by 2.9 times, and the plasma IGF-1 level decreased by 60% [38]. The decreased levels of testosterone, estrogen, and growth hormone, along with aging, play an important role in accelerating the decrease in muscle mass/strength and increase in fat mass [39]. Along with this, the expression of pyruvate dehydrogenase kinase 4 (PDK4), which contributes to pyruvate dehydrogenase phosphorylation and impaired glucose utilization, also increased 2–4 fold in skeletal muscles after a high-fat diet/ketogenic diet [38], indicating that muscle tissue after high-fat diet/ketogenic diet shifted the preferred energy substrate from glucose to fat, and other studies repeatedly confirmed the increase of PDK4 content in skeletal muscle after a high-fat diet/ketogenic diet [40,41]. In addition, mice fed with a ketogenic diet could cause the upregulation of muscle atrophy-related genes *Mafbx*, *Murf1*, *Foxo3*, *Lc3b*, and *Klf15* in skeletal muscle, and the expression of anabolic genes *Igf1* and *Col1a2* decreased in ketogenic diet-fed mice [38]. Furthermore, our previous large cross-sectional study showed that high fat intake was associated with decreased sleep duration [42], and children with short sleep durations (<8 h/day) tended to have higher BMI values than those with normal sleep durations (8–9 h/day) [43].

### 4.3. Strengths and Limitations

The current study has several notable strengths, including its large sample size and use of well-validated and precise techniques for measuring body composition. However, the study also has several limitations that must be acknowledged. Firstly, the cross-sectional design of NHANES makes it challenging to elucidate a causal relationship between macronutrients intake distribution and body composition. Secondly, we only analyzed the macronutrient intake based on the first 24-h dietary recall as the data were based on face-to-face interviews. The second 24-h dietary data were not used as a telephone interview was conducted. The dietary recall accuracy may be compromised due to recall bias and the Hawthorne effect. Lastly, to further evaluate the effects of macronutrients on body composition, prospective studies with rigorous design and mobile application for self-monitoring dietary intake are needed.

In conclusion, the results from the current study indicated that a higher percentage of energy intake from carbohydrate was associated with lower fat mass and lower prevalence of high fat, while higher fat consumption was significantly associated with lower muscle mass, higher fat mass and high risk of SO. Replacing carbohydrate intake with fat intake was associated with SO. The change in children’s diet towards a healthy diet with low fat composition can help prevent sarcopenic obesity. However, more studies are needed to further validate our findings.

## Figures and Tables

**Figure 1 nutrients-15-02307-f001:**
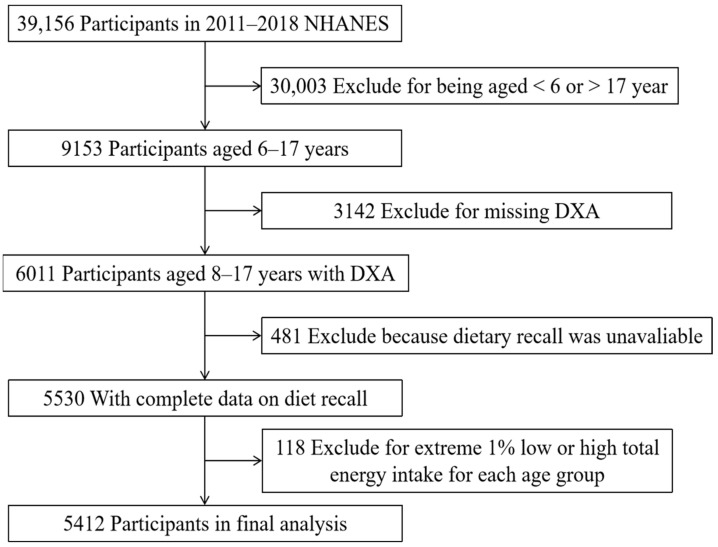
Flowchart of the sample selection from NHANES 2011–2018.

**Table 1 nutrients-15-02307-t001:** Sample characteristics by sarcopenia and obesity status among children and adolescents attending NHANES 2011–2018 (N = 5412).

	Total	Normal (N = 3816)	Obesity Alone (N = 156)	Sarcopenia Alone (N = 596)	Sarcopenic Obesity (N = 844)	*p*
Age (years)	12.3 ± 2.9	12.4 ± 2.9	13.0 ± 2.7	11.8 ± 2.6	12.2 ± 2.9	<0.001
Gender						<0.001
Boys	2780 (51.4)	1928 (50.5)	67 (42.9)	351 (58.9)	434 (51.4)	
Girls	2632 (48.6)	1888 (49.5)	89 (57.1)	245 (41.1)	410 (48.6)	
Race/ethnicity						<0.001
Non-Hispanic						
White	1486 (27.5)	1090 (28.6)	28 (17.9)	160 (26.8)	208 (24.6)	
Black	1320 (24.4)	974 (25.5)	55 (35.3)	69 (11.6)	222 (26.3)	
Mexican American	1152 (21.3)	700 (18.3)	40 (25.6)	186 (31.2)	226 (26.8)	
Other Hispanic	569 (10.5)	378 (9.9)	17 (10.9)	84 (14.1)	90 (10.7)	
Other Race	885 (16.3)	674 (17.7)	16 (10.3)	97 (16.3)	98 (11.6)	
BMI (kg/m^2^)	21.9 ± 5.8	19.5 ± 3.3	28.9 ± 3.2	22.8 ± 3.0	31.2 ± 5.7	<0.001
MET minutes per week	827.1 ± 748.2	860.8 ± 761.9	851.3 ± 740.3	637.6 ± 591.7	755.8 ± 742.4	<0.001
Body composition						
Muscle mass (kg)	35.2 ± 12.0	33.4 ± 11.3	49.9 ± 14.0	31.4 ± 8.4	43.5 ± 13.8	<0.001
Fat mass (kg)	16.8 ± 9.8	12.5 ± 5.1	25.1 ± 5.6	20.2 ± 6.2	32.3 ± 10.9	<0.001
Total energy intake (kcal/d)	1963.9 ± 779.9	1995.7 ± 790.9	2007.2 ± 895.5	1883.4 ± 675.4	1869.2 ± 765.5	<0.001
Dietary fiber intake (g/d)	14.4 ± 7.9	14.5 ± 8.1	14.0 ± 7.8	14.2 ± 7.7	13.9 ± 7.7	0.223
Macronutrients intake						
Protein intake (%E)	14.3 ± 4.7	14.2 ± 4.6	14.5 ± 5	14.6 ± 5	14.8 ± 4.8	0.052
Carbohydrate intake (%E)	52.1 ± 9.2	52.3 ± 9.1	51.3 ± 9	51.5 ± 9.4	51.0 ± 9.6	0.085
Fat intake (%E)	33.6 ± 7.8	33.5 ± 7.7	34.2 ± 8	33.9 ± 7.7	34.4 ± 7.9	0.008

Mean ± standard deviation for continuous variables and *p* value was calculated by ANOVA. N (%) for categorical variables and *p* value was calculated by weighted chi-square test. BMI, body mass index; MET, metabolic equivalent task.

**Table 2 nutrients-15-02307-t002:** The association of macronutrient intake with muscle mass and fat mass among 5412 children and adolescents from 2011–2018 NHANES.

	Muscle Mass		Fat Mass	
	Crude	Model 1	Crude	Model 1
	Muscle mass index Z-score, β (95% CI)		Fat mass index Z-score, β (95% CI)	
Protein intake (5% increase)	0.32 (0.24, 0.40) ***	0.01 (−0.03, 0.06)	0.23 (0.12, 0.33) ***	−0.01 (−0.06, 0.04)
Carbohydrate intake (5% increase)	−0.09 (−0.13, −0.05) ***	0.02 (−0.01, 0.04)	−0.06 (−0.11, −0.01) *	−0.02 (−0.04, −0.01) *
Fat intake (5% increase)	0.01 (−0.04, 0.05)	−0.03 (−0.06, −0.01) *	0.01 (−0.05, 0.07)	0.03 (0.01, 0.06) *
	Low muscle mass, OR (95% CI)		High fat mass, OR (95% CI)	
Protein intake (5% increase)	0.96 (0.88, 1.05)	1.02 (0.85, 1.23)	1.18 (1.10, 1.27) *	1.19 (0.91, 1.54)
Carbohydrate intake (5% increase)	1.01 (0.96, 1.05)	1.00 (0.91, 1.09)	0.96 (0.92, 0.99) **	0.88 (0.76, 0.99) *
Fat intake (5% increase)	1.00 (0.95, 1.05)	1.01 (0.90, 1.11)	1.17 (1.01, 1.28) ***	1.13 (1.02, 1.35) *

Crude model: no covariate was adjusted. Model 1: age, gender, metabolic equivalent task minutes, dietary fiber intake and total energy intake were adjusted. Fat mass index and muscle mass index were age- and sex-specific z-score transformed. CI: confidence interval, OR: odds ratio. *: 0.01 < *p* < 0.05, **: 0.001 < *p* < 0.01, ***: *p* < 0.001.

**Table 3 nutrients-15-02307-t003:** Association of macronutrients intake as a percentage of energy intake with sarcopenia and obesity status among 5412 children and adolescents attending 2011–2018 NHANES.

Sarcopenia and Obesity Status	Protein Intake (5 %E)		Carbohydrate Intake (5 %E)		Fat Intake (5 %E)	
	Crude	Model 1	Crude	Model 1	Crude	Model 1
Normal	Ref	Ref	Ref	Ref	Ref	Ref
Obesity alone	1.06 (0.95, 1.17)	1.04 (0.98, 1.09)	0.95 (0.87, 1.03)	0.95 (0.91, 0.99) *	1.07 (0.90, 1.26)	1.10 (1.01, 1.20) *
Sarcopenia alone	0.57 (0.26, 1.27)	0.99 (0.82, 1.18)	0.76 (0.44, 1.32)	0.90 (0.82, 1.10)	3.69 (1.18, 5.48) *	3.54 (1.14, 5.08) *
Sarcopenic obesity	0.61 (0.01, 1.33)	0.85 (0.64, 1.13)	0.52 (0.03, 1.83)	0.93 (0.80, 1.08)	4.38 (1.11, 6.44) *	4.42 (1.47, 6.59) *

Crude model: no covariate was adjusted. Model 1: age, gender, metabolic equivalent task minutes, dietary fiber intake and total energy intake were adjusted. *: *p* < 0.05.

**Table 4 nutrients-15-02307-t004:** Association of isocaloric substitution of macronutrients (5 %E) with muscle mass and fat mass among 5412 children and adolescents from 2011–2018 NHANES.

Isocaloric Substitution (5 %E)	Muscle Mass		Fat Mass	
	Muscle Mass Index Z-Score, β (95% CI)	Low Muscle Mass, OR (95% CI)	Fat Mass Index Z-Score, β (95% CI)	High Fat Mass, OR (95% CI)
Carbohydrate substituting protein	−0.02 (−0.06, 0.03)	0.99 (0.82, 1.20)	0.01 (−0.04, 0.06)	0.81 (0.62, 1.06)
Fat substituting protein	−0.05 (−0.10, 0.01)	0.99 (0.80, 1.24)	0.04 (−0.02, 0.10)	0.91 (0.66, 1.26)
Fat substituting carbohydrate	−0.03 (−0.06, −0.01) *	0.99 (0.90, 1.12)	0.03 (0.01, 0.06) *	1.12 (1.04, 1.34) *

Model was adjusted for age, gender, metabolic equivalent task minutes, and total energy intake. Macronutrients intakes also entered multivariate regression models apart from the substituted one. Fat mass index and muscle mass index were age- and sex-specific z-score transformed. CI: confidence interval, OR: odds ratio. *: *p* < 0.05.

**Table 5 nutrients-15-02307-t005:** Association of isocaloric substitution of macronutrients (5 %E) with sarcopenia and obesity status among 5412 children and adolescents attending 2011–2018 NHANES.

Sarcopenia and Obesity Status	Carbohydrate Substituting Protein	Fat Substituting Protein	Fat Substituting Carbohydrate
Normal	Ref	Ref	Ref
obesity alone	2.26 (0.81, 6.25)	2.25 (1.53, 4.40) *	3.66 (1.19, 5.24) *
Sarcopenia alone	1.03 (0.86, 1.24)	1.21 (1.07, 1.52) *	1.17 (1.04, 1.32) *
Sarcopenic obesity	1.08 (0.65, 4.98)	2.36 (1.18, 3.18) *	3.54 (1.15, 5.87) *

The model was adjusted for age, gender, metabolic equivalent task minutes, and total energy intake. Macronutrients intakes also entered multivariate regression models apart from the substituted one. *: *p* < 0.05.

## Data Availability

The datasets generated and analyzed during the present study are available in the NHANES repository at https://www.cdc.gov/nchs/nhanes/index.htm (accessed on 20 November 2022).

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
