# Peer review of "Association of Macronutrients Intake with Body Composition and Sarcopenic Obesity in Children and Adolescents: A Population-Based Analysis of the National Health and Nutrition Examination Survey (NHANES) 2011–2018"

_nutrients, 2023, doi:10.3390/nu15102307_

Round 1

Reviewer 1 Report

This point will be needed to consider before proceed.

1) Are there any other study limitations?

Author Response

Point 1: Are there any other study limitations?

Response 1: Done. We revised and added the limitations as follows: Secondly, we only analyzed the macronutrient intake based on the first 24-hour dietary recall as the data were based on face-to-face interview. The the second 24-hour dietary data were not used as telephone interview was used. The dietary recall accuracy may be compromised due to recall bias and Hawthorne effect. Lastly, to further evaluate the effects of macronutrients on body composition, prospective studies with rigorous design and mobile application for self-monitoring dietary intake are needed.

Reviewer 2 Report

In this manuscript by Bing Yang et al., the authors argued that macronutrient intake is associated with sarcopenic obesity in children and adolescents. Based on a large-scale analysis of body composition information, the potential correlation between macromolecule intake, body composition, and sarcopenic obesity is intriguing. However, functional and mechanistic studies are weak in supporting their conclusions if they claim that a high-fat diet significantly increases sarcopenic obesity in younger age groups. Perhaps this manuscript could be improved with additional data and discussion insights on the effects of high-fat diets on sarcopenic obesity in children and adolescents.

Major comments

1. According to the current manuscript, macronutrients or macromolecules like to include three types of food composition: proteins, carbohydrates, and fats. Therefore, the authors should clarify which nutrient intake is more important to make definitive conclusion. In the title/conclusion, macronutrient or macromolecular intake is not specific for definitive conclusions.

2. Authors should address whether high-fat diets affect sarcopenic obesity only in younger ages and teenagers, but not adults. Is sarcopenic obesity caused by a fat diet independent of age?

3. Although the mechanism of sarcopenic obesity caused by a high-fat diet has been explained in several sentences in the Discussion section, it is necessary to explain why a high-fat/ketogenic diet is more important for the sarcopenic phenotype. In addition to reference #35 paper, it would be nice if you could find and explain more functional studies references related to specific genes or clinical evidence.

Author Response

Point 1: In this manuscript by Bing Yang et al., the authors argued that macronutrient intake is associated with sarcopenic obesity in children and adolescents. Based on a large-scale analysis of body composition information, the potential correlation between macromolecule intake, body composition, and sarcopenic obesity is intriguing. However, functional and mechanistic studies are weak in supporting their conclusions if they claim that a high-fat diet significantly increases sarcopenic obesity in younger age groups. Perhaps this manuscript could be improved with additional data and discussion insights on the effects of high-fat diets on sarcopenic obesity in children and adolescents.

Response 1: Thanks for your comments. We have further discussed the mechanism of a high-fat diet on sarcopenic obesity by adding additional previous research data as follows: And the decreased levels of testosterone, estrogen, and growth hormone with aging play an important role in accelerating the decrease in muscle mass/strength and increase in fat mass [1]. Along with this, the expression of pyruvate dehydrogenase kinase 4 (PDK4), which contributed to pyruvate dehydrogenase phosphorylation and impaired glucose utilization, increased 2-4 fold in skeletal muscles after high-fat diet/ketogenic diet [2], indicating that muscle tissue after high-fat diet/ketogenic diet shifted the preferred energy substrate from glucose to fat, and other studies repeatedly confirmed the increase of PDK4 content in skeletal muscle after a high-fat diet/ketogenic diet [3,4]. 

References: [1] Vincent, H.K.; Raiser, S.N.; Vincent, K.R. The aging musculoskeletal system and obesity‐related considerations with exercise. Ageing Res. Rev. 2012, 11: 361-373. [2] Nakao, R.; Abe, T.; Yamamoto, S.; Oishi, K. Ketogenic diet induces skeletal muscle atrophy via reducing muscle protein synthesis and possibly activating proteolysis in mice. Sci. Rep. 2019, 9, 19652. [3] Rinnankoski-Tuikka, R.; Silvennoinen, M.; Torvinen, S.; Hulmi, J.J.; Lehti, M.; Kivelä, R.; Reunanen, H.; Kainulainen, H. Effects of high-fat diet and physical activity on pyruvate dehydrogenase kinase-4 in mouse skeletal muscle. Nutr. Metab. 2012, 9, 53. [4] Wende, A.R.; Huss, J.M.; Schaeffer, P.J.; Giguère, V.; Kelly, D.P. PGC-1alpha coactivates PDK4 gene expression via the orphan nuclear receptor ERRalpha: A mechanism for transcriptional control of muscle glucose metabolism. Mol. Cell. Biol. 2005, 25, 10684-10694.

Point 2: According to the current manuscript, macronutrients or macromolecules like to include three types of food composition: proteins, carbohydrates, and fats. Therefore, the authors should clarify which nutrient intake is more important to make definitive conclusion. In the title/conclusion, macronutrient or macromolecular intake is not specific for definitive conclusions.

Response 2: Thanks for your comments. All three macronutrients are important to our body. This study found that a high-fat diet coupled with low carbohydrate/protein intake are associated with sarcopenic obesity among children and adolescents. Based on your comments and the results, the conclusion have been revised as follows: In conclusion, a high-fat diet coupled with low carbohydrate/protein intake are associated with sarcopenic obesity among children and adolescents. The change in children's diet towards a healthy diet with low fat composition may help prevent sarcopenic obesity. However, randomized clinical trials or longitudinal studies are needed to further validate our findings.

Point 3: Authors should address whether high-fat diets affect sarcopenic obesity only in younger ages and teenagers, but not adults. Is sarcopenic obesity caused by a fat diet independent of age?

Response 3: Thanks for your suggestion. A high-fat diet not only affects sarcopenic obesity in children and adolescents, but also affects adults. And prior research on this subject has been largely limited to small sample sizes or adult populations. This study further examined the association between macronutrients and body composition, as well as sarcopenic obesity in a large sample size of children and adolescents, and compared the results between children and adults in the Introduction and Discussion sections.

Point 4: Although the mechanism of sarcopenic obesity caused by a high-fat diet has been explained in several sentences in the Discussion section, it is necessary to explain why a high-fat/ketogenic diet is more important for the sarcopenic phenotype. In addition to reference #35 paper, it would be nice if you could find and explain more functional studies references related to specific genes or clinical evidence.

Response 4: Thanks for your comments. We have discussed more functional studies with references as follows: Along with this, the expression of pyruvate dehydrogenase kinase 4 (PDK4), which contributed to pyruvate dehydrogenase phosphorylation and impaired glucose utilization, increased 2-4 fold in skeletal muscles after high-fat diet/ketogenic diet [1], indicating that muscle tissue after high-fat diet/ketogenic diet shifted the preferred energy substrate from glucose to fat, and other studies repeatedly confirmed the increase of PDK4 content in skeletal muscle after a high-fat diet/ketogenic diet [2,3]. 

References: [1] Nakao, R.; Abe, T.; Yamamoto, S.; Oishi, K. Ketogenic diet induces skeletal muscle atrophy via reducing muscle protein synthesis and possibly activating proteolysis in mice. Sci. Rep. 2019, 9, 19652. [2] Rinnankoski-Tuikka, R.; Silvennoinen, M.; Torvinen, S.; Hulmi, J.J.; Lehti, M.; Kivelä, R.; Reunanen, H.; Kainulainen, H. Effects of high-fat diet and physical activity on pyruvate dehydrogenase kinase-4 in mouse skeletal muscle. Nutr. Metab. 2012, 9, 53. [3] Wende, A.R.; Huss, J.M.; Schaeffer, P.J.; Giguère, V.; Kelly, D.P. PGC-1alpha coactivates PDK4 gene expression via the orphan nuclear receptor ERRalpha: A mechanism for transcriptional control of muscle glucose metabolism. Mol. Cell. Biol. 2005, 25, 10684-10694.

Reviewer 3 Report

1.       I suggest that NHANES should be written as a complete term, with the abbreviation in parenthesis.

21. Please, check for typo “5 E%”

23 Please explain clearly the meaning of “β, -0.03 [95%CI -0.06 to -0.01”

23-24. The results would be more interesting to be present differently, for example, showing the differences between the samples as percentages instead of CI.

25. In conclusion, information about protein should also be included.

31. You can avoid using parenthesis and include “SO” as a part of the text.

72. Delete “the”

93. Please, use the complete term of DXA

93. It would be important to add the equations as a formatted text in Word.

 109. “Participants,”

115. Was it necessary to include alcohol (Alcohol =4 kcal).

121. Is this calculation precise? You must specify what type of activities are categorized at these METs.

127. Use another title to describe the paragraph.

137. Don’t use the full term of SO again.

162. “shows”

167. Not in the Obesity group?

173. Does the P value reveal a statistically significant difference between measurement? If yes, did you conduct pairwise comparisons?

177. Probably, you mean “indicators”.

201. Show “p” in that format.

207. “Obesity alone”

292. You need to reference the literature to support this statement.

301. What about the physical activity? Are there any aspects of its contribution to the samples’ values? Was it a limitation for their participation in the study? 

Author Response

Point 1: Line 1: I suggest that NHANES should be written as a complete term, with the abbreviation in parenthesis.

Response 1: Done.

Point 2: Line 21: Please, check for typo “5 E%”

Response 2: Thanks. Done.

Point 3: Line 23: Please explain clearly the meaning of “β, -0.03 [95%CI -0.06 to -0.01”

Response 3: Done. We revised the sentences as follows: Substitute of carbohydrate (5 %E) with fat decreased muscle mass by 0.03 (95% CI 0.01 to 0.06), but increased fat mass by 0.03 (95% CI 0.01 to 0.06) and increased the prevalence of sarcopenic obesity by 254% (95% CI 15% to 487%).

Point 4: Line 23-24: The results would be more interesting to be present differently, for example, showing the differences between the samples as percentages instead of CI.

Response 4: Thanks. We explained the results using percentages.

Point 5: Line 25: In conclusion, information about protein should also be included.

Response 5: Done. We added protein information in the conclusion as follows: a high-fat diet coupled with low carbohydrate/protein intake are associated with sarcopenic obesity among children and adolescents.

Point 6: Line 31: You can avoid using parenthesis and include “SO” as a part of the text.

Response 6: Thanks. Done.

Point 7: Line 72: Delete “the”

Response 7: Done.

Point 8: Line 93: Please, use the complete term of DXA

Response 8: Done. The complete term was used on the Line 88 where DXA first appeared.

Point 9: Line 93: It would be important to add the equations as a formatted text in Word.

Response 9: Done. We changed the text to equations.

Point 10: Line 109: “Participants,”

Response 10: Thanks, we revised the sentence.

Point 11: Line 115: Was it necessary to include alcohol (Alcohol =4 kcal).

Response 11: Thank you for your suggestion. There was no need to include alcohol in the analysis, as this study was conducted among children and adolescents who are legally prohibited from drinking alcohol before the age of 21. And in this study, the variable of total energy intake was added to the model analysis to exclude the influence of other confounding factors on the results.

Point 12: Line 121: Is this calculation precise? You must specify what type of activities are categorized at these METs.

Response 12: Thanks for your question. The NHANES physical activity questionnaire (prefix PAQ) is based on the Global Physical Activity Questionnaire (GPAQ) [1], which assesses time spent on sedentary and typical physical activity over the past week. Questions captured time spent doing physical activity in various domains and by intensity, including vigorous and moderate activity at work, transport activity, and vigorous and moderate activity during leisure time. And we calculated metabolic equivalent of task (MET) minutes using NHANES-recommended conversions. This method was widely used by other research papers [2]. 

References: [1] Armstrong, T.; Bull, F. Development of the world health organization global physical activity questionnaire (GPAQ). Journal of Public Health. 2006, 14(2): 66–70. [2] Gao, S.; Qian, X.; Huang, S.; Deng, W.; Li, Z.; Hu, Y. Association between macronutrients intake distribution and bone mineral density. Clin Nutr. 2022, 41(8): 1689-1696.

Point 13: Line 127: Use another title to describe the paragraph.

Response 13: Done. We revised the title as follows: Definitions of sarcopenia and obesity.

Point 14: Line 137: Don’t use the full term of SO again.

Response 14: Deleted.

Point 15: Line 162: “shows”

Response 15: Thanks, Changed.

Point 16: Line 167: Not in the Obesity group?

Response 16: Thanks, it was in Obesity group, we revised it.

Point 17: Line 173: Does the P value reveal a statistically significant difference between measurement? If yes, did you conduct pairwise comparisons?

Response 17: Yes. This is a comparison of the differences between measurement, and we conducted pairwise comparison, but it was not shown in the Table 1. As it is only a description of sample characteristics, we would like not to expand on the comparisons.

Point 18: Line 177: Probably, you mean “indicators”.

Response 18: Yes, Thanks. It has been changed.

Point 19: Line 201: Show “p” in that format.

Response 19: Done.

Point 20: Line 207: “Obesity alone”

Response 20: Corrected, thank you.

Point 21: Line 292: You need to reference the literature to support this statement.

Response 21: Thanks, references have been listed and added in manuscript.

References: [1] Nakao, R.; Abe, T.; Yamamoto, S.; Oishi, K. Ketogenic diet induces skeletal muscle atrophy via reducing muscle protein synthesis and possibly activating proteolysis in mice. Sci. Rep. 2019, 9, 19652. [2] Vincent, H.K.; Raiser, S.N.; Vincent, K.R. The aging musculoskeletal system and obesity‐related considerations with exercise. Ageing Res. Rev. 2012, 11: 361-373. [3] Rinnankoski-Tuikka, R.; Silvennoinen, M.; Torvinen, S.; Hulmi, J.J.; Lehti, M.; Kivelä, R.; Reunanen, H.; Kainulainen, H. Effects of high-fat diet and physical activity on pyruvate dehydrogenase kinase-4 in mouse skeletal muscle. Nutr. Metab. 2012, 9, 53. [4] Wende, A.R.; Huss, J.M.; Schaeffer, P.J.; Giguère, V.; Kelly, D.P. PGC-1alpha coactivates PDK4 gene expression via the orphan nuclear receptor ERRalpha: A mechanism for transcriptional control of muscle glucose metabolism. Mol. Cell. Biol. 2005, 25, 10684-10694. [5] Shi, Z.; McEvoy, M.; Luu, J.; Attia, J. Dietary fat and sleep duration in Chinese men and women. Int. J. Obes. (Lond) 2008, 32(12), 1835-40. [6] Gao, L.; Wu, Y.; Zhu, J.; Wang, W.; Wang, Y. Associations of sleep duration with childhood obesity: findings from a national cohort study in China. Global Health Journal. 2022, 6, 149-155.

Point 22: Line 301: What about the physical activity? Are there any aspects of its contribution to the samples’ values? Was it a limitation for their participation in the study?

Response 22: Thanks for your questions. Physical activity is evaluated using fully validated questionnaires and collected by trained personnel, and we calculated metabolic equivalent of task (MET) minutes using NHANES-recommended conversions. Based on the above information, the inclusion of physical activity variables in this study is accurate and complete, and may not be considered a limitation of this study.

The NHANES physical activity questionnaire (prefix PAQ) is based on the Global Physical Activity Questionnaire (GPAQ) [1], which assesses time spent sitting and time spent engaged in typical physical activity over the past week. Questions captured time spent doing physical activity in various domains and by intensity, including vigorous and moderate activity at work, transport activity, and vigorous and moderate activity during leisure time.

Reference: [1] Armstrong, T.; Bull, F. Development of the world health organization global physical activity questionnaire (GPAQ). Journal of Public Health. 2006, 14(2): 66–70.

Reviewer 4 Report

Concerns I have the following suggestions for the authors to consider: Major 1. Methodology-Assessment of macronutrients intake distribution: The authors mentioned that “the nutrient intake obtained from food was estimated using two non-consecutive 24-hour dietary recalls”, however “the recalls were chosen from the first day” (p3, ln105-106, ln113-114). Could the authors justify why they chose to only use the data from only the first day, if they could ideally have received a better estimate of each participant’s nutrient intake if they took an average from both days, or more days? The authors could benefit from elaborating on why they chose to estimate the nutrient intake from the first day of 24h dietary recall instead of both days, and its effect on sample size and accuracy of face-to-face survey data. Additionally, the authors could justify why two non-consecutive 24h dietary recalls were conducted if only one was necessary. Minor 1. Recommend more specific wording than “promoting healthy eating is needed to prevent sarcopenic obesity”, as this study isn’t focused on the benefits of healthy eating. A more appropriate statement would be something that highlights the detriments of high fat intake and the need to address that (p1, ln27). Furthermore, this need to promote healthy eating is not elaborated in the rest of the paper. 2. Methodology: The macronutrients intake of the participants was obtained through a 24h dietary recall method. Were the participants aware that they would have to report their dietary intake prior to the survey? 3. Strengths & Limitations: Have the authors considered the Hawthorne effect affecting the reported dietary intake, as a possible limitation? In addition, the accuracy could be undermined by memory lapses and incorrect quantification in the 24h dietary recall. 4. In “substituting fat with carbohydrates can lead to a decrease in muscle mass” (p7, ln259), did the authors mean to write “substituting carbohydrates with fat can lead to a decrease in muscle mass” 5. Recommend deletion “may does not reflect the usual intake” (p8, ln309) 6. Formatting for Table 3 (p6), recommend expanding the table so the brackets lines up. 7. Formatting of (p8, ln277) the text on this line looks condensed, might be a different font. 8. Recommend adding more references to strengthen evidence that their finding aligns with previous studies (p8, ln287-288) 9. Spelling error identified “including its large sample size” (p8, ln302) 10. Recommend addition “while higher fat consumption” (p8, ln312) 11. Authors could benefit from adding a section following the discussion section elaborating on the implications of their findings and how this links back to their statement “promoting healthy eating is needed to prevent sarcopenic obesity”. 12. Recommend adding “These findings” (p8, ln280), “mice fed with a ketogenic diet” (p8, ln294), and “that a higher percentage” (p8, ln310) 13. Limitations & Further studies: The authors could also include the further studies they would recommend as a follow up to their statement in (p9, ln314), such as experimental designs that would address their limitations e.g. mobile application for self-monitoring dietary intake. 14. To emphasise on how this study addresses the gap in current knowledge, the authors could benefit from highlighting that this study is conducted on children and adolescents, and the significance of these findings in contrast to prior studies conducted on adult populations.

Author Response

Major

Point 1: Methodology-Assessment of macronutrients intake distribution: The authors mentioned that “the nutrient intake obtained from food was estimated using two non-consecutive 24-hour dietary recalls”, however “the recalls were chosen from the first day” (p3, ln105-106, ln113-114). Could the authors justify why they chose to only use the data from only the first day, if they could ideally have received a better estimate of each participant’s nutrient intake if they took an average from both days, or more days? The authors could benefit from elaborating on why they chose to estimate the nutrient intake from the first day of 24h dietary recall instead of both days, and its effect on sample size and accuracy of face-to-face survey data. Additionally, the authors could justify why two non-consecutive 24h dietary recalls were conducted if only one was necessary.

Response 1: Thanks for your comments. Two interviews of 24-h dietary recall were adopted, and all participants underwent in-person interviews for the initial 24-h dietary recall interview. Then, 3-10 days later, a portion of the children and adult participants took part in a second, telephone-based 24-h dietary recall interview [1]. Indeed, two 24-h dietary recall can make dietary data more accurate, but it can lead to a missing sample size, as the second survey was not conducted for the entire population in the first survey, and face-to-face surveys used physical utensils, while telephone surveys used images distributed to respondents. In addition, previous adult studies have only used data from the first survey, which is true and reliable, but there may be limitation to assess macronutrients intake fluctuation [2].

In this study, we modified the section about “assessment of macronutrients intake distribution” to clarify the evaluation method, and we also elucidated the limitations of using single dietary recall data as follows: Secondly, we only analyzed the macronutrient intake based on the first 24-hour dietary recall as the data were based on face-to-face interview. The the second 24-hour dietary data were not used as telephone interview was used. The dietary recall accuracy may be compromised due to recall bias and Hawthorne effect.

References: [1] Zipf, G.; Chiappa, M.; Porter, K.; Ostchega, Y.; Lewis, B.; Dostal, J. National health and nutrition examination survey: plan and operations, 1999-2010. Vital Health Stat 1. 2013, (56): 1-37. [2] Gao, S.; Qian, X.; Huang, S.; Deng, W.; Li, Z.; Hu, Y. Association between macronutrients intake distribution and bone mineral density. Clin Nutr. 2022, 41(8): 1689-1696.

Minor

Point 1: Recommend more specific wording than “promoting healthy eating is needed to prevent sarcopenic obesity”, as this study isn’t focused on the benefits of healthy eating. A more appropriate statement would be something that highlights the detriments of high fat intake and the need to address that (p1, ln27). Furthermore, this need to promote healthy eating is not elaborated in the rest of the paper.

Response 1: Thanks for your comments. We revised the conclusion as follows: In conclusion, a high-fat diet coupled with low carbohydrate/protein intake are associated with sarcopenic obesity among children and adolescents. The change in children's diet towards a healthy diet with low fat composition may help prevent sarcopenic obesity. However, randomized clinical trials or longitudinal studies are needed to further validate our findings.

Point 2: Methodology: The macronutrients intake of the participants was obtained through a 24h dietary recall method. Were the participants aware that they would have to report their dietary intake prior to the survey?

Response 2: Participants were informed of the survey process and their rights as a participant by interviewers and written materials (such as brochures and flyers) before the survey process begins, but he or she would not have known about the investigation process earlier. After the participant understood the entire NHANES process, he or she had the opportunity to consent or assent to participate.

Point 3: Strengths & Limitations: Have the authors considered the Hawthorne effect affecting the reported dietary intake, as a possible limitation? In addition, the accuracy could be undermined by memory lapses and incorrect quantification in the 24h dietary recall.

Response 3: Thanks for your comments. Yes, we agree that Hawthorne effect may affect the reported dietary intake, although participants will not be aware of the specific investigation process in advance. Based on your comments, we have revised the Limitations as follows: Secondly, we only analyzed the macronutrient intake based on the first 24-hour dietary recall as the data were based on face-to-face interview. The the second 24-hour dietary data were not used as telephone interview was used. The dietary recall accuracy may be compromised due to recall bias and Hawthorne effect.

Point 4: In “substituting fat with carbohydrates can lead to a decrease in muscle mass” (p7, ln259), did the authors mean to write “substituting carbohydrates with fat can lead to a decrease in muscle mass”

Response 4: Yes, we revised.

Point 5: Recommend deletion “may does not reflect the usual intake” (p8, ln309)

Response 5: Done.

Point 6: Formatting for Table 3 (p6), recommend expanding the table so the brackets lines up.

Response 6: Thanks. The brackets in tables on our computer were aligned, and different computer or software versions might cause formatting errors. We would ensure that the final version was formatted correctly.

Point 7: Formatting of (p8, ln277) the text on this line looks condensed, might be a different font.

Response 7: Thanks. We checked and it was correct.

Point 8: Recommend adding more references to strengthen evidence that their finding aligns with previous studies (p8, ln287-288)

Response 8: Done. We added the references and revised the sentences as follows: Regarding muscles, we did not find any effects of carbohydrate intake, which is consistent with the previous study in 3573 children aged 1-10 years [1]. In addition, a randomized controlled trial in adults aged 18 to 41 also did not find that intake of carbohydrates is associated with muscle mass [2].

References: [1] Nguyen, A.N.; Santos, S.; Braun, K.; Voortman, T. Carbohydrate Intake in Early Childhood and Body Composition and Metabolic Health: Results from the Generation R Study. Nutrients 2020, 12(7), 1940. [2] Wachsmuth, N.B.; Aberer, F.; Haupt, S.; Schierbauer, J.R.; Zimmer, R.T.; Eckstein, M.L.; Zunner, B.; Schmidt, W.; Niedrist, T.; Sourij, H.; et al. The Impact of a High-Carbohydrate/Low Fat vs. Low-Carbohydrate Diet on Performance and Body Composition in Physically Active Adults: A Cross-Over Controlled Trial. Nutrients. 2022, 14(3): 423.

Point 9: Spelling error identified “including its large sample size” (p8, ln302)

Response 9: Thanks. We have revised the word.

Point 10: Recommend addition “while higher fat consumption” (p8, ln312)

Response 10: Done. Thanks.

Point 11: Authors could benefit from adding a section following the discussion section elaborating on the implications of their findings and how this links back to their statement “promoting healthy eating is needed to prevent sarcopenic obesity”.

Response 11: Thanks for your comments. We added sentences in Discussion section as follows: The change in children's diet towards a healthy diet with low fat composition can help prevent sarcopenic obesity. And we revised the sentence about “promoting healthy eating is needed to prevent sarcopenic obesity” in Abstract section based on your previous suggestion (Please see Minor Ponit 1).

Point 12: Recommend adding “These findings” (p8, ln280), “mice fed with a ketogenic diet” (p8, ln294), and “that a higher percentage” (p8, ln310)

Response 12: Done.

Point 13: Limitations & Further studies: The authors could also include the further studies they would recommend as a follow up to their statement in (p9, ln314), such as experimental designs that would address their limitations e.g. mobile application for self-monitoring dietary intake.

Response 13: Thanks for your constructive suggestion. We revised the Limitations & Further studies as follows: Lastly, to further evaluate the effect of macronutrients on body composition, prospective studies with rigorous design and mobile application for self-monitoring dietary intake are needed.

Point 14: To emphasise on how this study addresses the gap in current knowledge, the authors could benefit from highlighting that this study is conducted on children and adolescents, and the significance of these findings in contrast to prior studies conducted on adult populations.

Response 14: Thanks for your suggestion. We emphasized the highlights in Discussion section as follows: And our study was conducted among children and adolescents, whose body composition is rapidly developing, especially with the rapid accumulation of muscle and bone during childhood. Compared to adult studies, the results of this study also provides important evidence for the intervention and early prevention of unhealthy body composition in children.

Reviewer 5 Report

In the manuscript entitled "Association of macronutrients intake with body composition and sarcopenic obesity in children and adolescents: A Population-Based Analysis of NHANES 2011-2018 ", the author conducted a comprehensive examination of the relationship between the distribution of dietary macronutrients and parameters of the body composition by using data from National Health and Nutrition Examination Survey (NHANES) 2011-2018 and concluded that the high-fat diet substituting carbohydrate with fat are associated with sarcopenic obesity among children and adolescents in the United States. The finding is interesting and all the results are clearly presented.

Author Response

Point 1: In the manuscript entitled "Association of macronutrients intake with body composition and sarcopenic obesity in children and adolescents: A Population-Based Analysis of NHANES 2011-2018 ", the author conducted a comprehensive examination of the relationship between the distribution of dietary macronutrients and parameters of the body composition by using data from National Health and Nutrition Examination Survey (NHANES) 2011-2018 and concluded that the high-fat diet substituting carbohydrate with fat are associated with sarcopenic obesity among children and adolescents in the United States. The finding is interesting and all the results are clearly presented.

Response 1: Thanks for your comments.

Round 2

Reviewer 2 Report

The authors are faithfully updated. I agree to the publication on Nutrients.

Reviewer 3 Report

The authors explained and modified the manuscript according to the comments. 

Reviewer 4 Report

All good.